# Nucleus-Targeting Nanoplatform Based on Dendritic Peptide for Precise Photothermal Therapy

**DOI:** 10.3390/polym15071753

**Published:** 2023-03-31

**Authors:** Wen-Song Wang, Xiao-Yu Ma, Si-Yao Zheng, Si Chen, Jin-Xuan Fan, Fan Liu, Guo-Ping Yan

**Affiliations:** 1Hubei Key Laboratory of Plasma Chemistry and Advanced Materials, School of Material Science and Engineering, Wuhan Institute of Technology, Wuhan 430205, China; 2Key Laboratory of Biomedical Photonics (HUST), Ministry of Education, Huazhong University of Science and Technology, Wuhan 430074, China

**Keywords:** nuclear localization, precise photothermal therapy, arginine-rich dendritic peptide, tumor inhibition

## Abstract

Photothermal therapy directly acting on the nucleus is a potential anti-tumor treatment with higher killing efficiency. However, in practical applications, it is often difficult to achieve precise nuclear photothermal therapy because agents are difficult to accurately anchor to the nucleus. Therefore, it is urgent to develop a nanoheater that can accurately locate the nucleus. Here, we designed an amphiphilic arginine-rich dendritic peptide (RDP) with the sequence CRRK(RRCG(Fmoc))_2_, and prepared a nucleus-targeting nanoplatform RDP/I by encapsulating the photothermal agent IR780 in RDP for precise photothermal therapy of the tumor nucleus. The hydrophobic group Fmoc of the dendritic peptide provides strong hydrophobic force to firmly encapsulate IR780, which improves the solubility and stability of IR780. Moreover, the arginine-rich structure facilitates cellular uptake of RDP/I and endows it with the ability to quickly anchor to the nucleus. The nucleus-targeting nanoplatform RDP/I showed efficient nuclear enrichment ability and a significant tumor inhibition effect.

## 1. Introduction

As the center of cell genetics and metabolic regulation, the nucleus is one of the most important organelles in cells [1]. In the process of antitumor therapy, compared with other organelles, the destruction of the nucleus can more directly destroy the cell structure, interfere with cell metabolism, and even completely kill cells [2]. Many therapeutic strategies acting on the nucleus have achieved efficient cell killing by interfering with and destroying nuclear DNA, chromosomes, microtubule proteins, etc. [3,4,5]. Among them, photothermal therapy (PTT), as a new antitumor method, has the advantages of strong control ability, deep tissue penetration, high treatment efficiency, and weak side effects [6,7]. When PTT directly anchors to the nucleus, it can damage DNA by accurately nucleus-burning, thereby destroying the function and behavior of cells and achieving efficient cell killing ability [8,9]. More importantly, the precise destruction of the nucleus can effectively cutoff the triggering of inhibition mechanisms to photothermal effect, such as the up-regulation of intracellular heat shock protein, so as to ensure the efficient killing ability of tumor cells and inhibit the occurrence of metastasis [10,11]. Therefore, precise nucleus-targeting PTT is a promising cancer treatment strategy.

However, in order to achieve precise nuclear photothermal therapy, it is necessary to overcome the obstacles in the delivery process and successfully anchor the photothermal agent to the nucleus. Therefore, nanoplatforms must have good stability, excellent cell uptake ability, precise nucleus-targeting function, and superb photothermal conversion performance [12,13]. Near-infrared (NIR) indocyanine dyes show excellent performance in fluorescence imaging diagnosis and optical therapy. Among them, IR780 has superb NIR photothermal conversion performance and is a potential phototherapy agent [14,15]. Nevertheless, IR780 is heavily hindered for biological application by its limited solubility, poor stability, rapid metabolism, and indiscriminate toxicity [16,17]. Encapsulating IR780 with carriers can effectively overcome the above shortcomings. As a potential carrier with good biocompatibility, easy modification, and diverse biological functions, dendritic peptides have been widely used for agent encapsulation [18,19]. The interior wide cavity of dendritic peptides is conducive to the package of hydrophobic agents during self-assembly, while the external multi-arm structure of dendritic peptides is beneficial to functional group modification and networking. Therefore, dendritic peptides have shown unique advantages and potential in the encapsulation of agents [20]. Due to the presence of the guanidino group and the strong positive charge of arginine, dendritic peptides with arginine-rich sequences possess the admirable abilities of cell penetration and nuclear localization [21]. The encapsulation of IR780 by multifunctional dendritic peptides with arginine-rich sequences is not only expected to solve the above defects of IR780 but also endow it with the function of precise phototherapy to the nucleus.

In this study, we prepared the nanoheater RDP/I by encapsulating the photothermal agent IR780 in an arginine-rich dendritic peptide (RDP) with the sequence CRRK(RRCG(Fmoc))_2_. The encapsulation of an amphiphilic dendritic peptide could enhance the solubility and stability of IR780. Moreover, the arginine-rich structure of a peptide could endow membrane penetration ability and nuclear localization function to RDP/I. As shown in Figure 1, after accumulating in tumor tissues, nanoheater RDP/I was internalized in cells facilely owing to the membrane penetrating ability of the arginine-rich sequence. Then, nanoheater RDP/I was anchored to the nucleus rapidly through the nuclear localization ability of a peptide. Finally, under NIR irradiation, the precise nuclear photothermal effect of RDP/I led to efficient cell death. Experiments have confirmed that RDP/I can efficiently anchor to the nucleus and effectively kill tumor cells through precise nucleus-targeting phototherapy. This nanoplatform RDP/I provides a simple and effective strategy to customize the photothermal system for precise delivery of tumor nuclei.

## 2. Materials and Methods

### 2.1. Materials

The 2-chlorotrityl chloride resin, 1-hydroxybenzotriazole (HOBt) O-benzotriazole-N,N,N’,N’-tetramethyl-uronium-hexafluoro-phosphate (HBTU), and N-fluorenyl-9-methoxycarbonyl (Fmoc) protected L-amino acids (Fmoc-Cys(Trt)-OH, Fmoc-Arg(Pbf)-OH) and Fmoc-Lys(Fmoc)-OH) were obtained from GL Biochem. Ltd. Diisopropylethylamine, N,N-dimethylformamide (DMF), piperidine, anhydrous ether, thioanisole, ethanedithiol, trifluoroacetic acid, and dimethylsulfoxide (DMSO) were purchased from Shanghai Chemical Co. IR780 was obtained from Sigma-Aldrich LLC. All other reagents were analytical grade and used as received.

Fetal bovine serum (FBS), Roswell Park Memorial Institute medium (RPMI) 1640, minimum essential medium (MEM), penicillin-streptomycin, and phosphate-buffered saline (PBS) were obtained from Shanghai XP Biomed Ltd. Hoechst 33342 was purchased from Lonza Group Ltd. The 3-(4,5-dimethylthiazol-2-yl)-2,5-diphenyltetrazolium bromide (MTT) was obtained from Innochem Ltd. An apoptosis and necrosis assay kit and Calcein AM were obtained from Beyotime Biotechnology.

### 2.2. Cell Lines

The China Center for Type Culture Collection (CCTCC) offered NIH3T3 and CT26 cells. NIH3T3 and CT26 cells were cultured in MEM (containing 10% FBS and 1% penicillin-streptomycin) or RPMI 1640 (containing 10% FBS and 1% penicillin-streptomycin), respectively.

### 2.3. Synthesis of RDP

According to the sequence of CRRK(RRCG(Fmoc))_2_ (Figure 1A), amino acids were bonded to 2-chlorotrityl chloride resin by utilizing the standard Fmoc-based solid-phase synthesis technique [22]. After all amino acids were successfully bonded, the peptides were cleaved from the resin. The mixture was collected and precipitated in ether. Afterwards, the precipitate was collected and freeze-dried to obtain RDP. The molecular weight of RDP was assayed by electrospray ionization mass spectrometry (ESI-MS) (Bruker Compact, Bruker, Germany) and nuclear magnetic resonance spectroscopy (NMR) (400MR, Agilent, USA).

### 2.4. Determination of Critical Micelle Concentration (CMC)

CMC was detected by a fluorescence spectrophotometer (Cary Ecilpse, Agilent, CA, USA). 0.1 mL of pyrene solution (6 × 10^−6^ M in acetone) was added to containers and evaporated at room temperature for 12 h. Peptide solutions with different concentrations were added to the above containers. The mixture solutions were shaken at 37 °C for 2 h and left overnight. The fluorescence intensity ratio of the third and first electron vibration bands (I_3_/I_1_) was plotted according to the excitation spectrum of pyrene and the logarithm of peptide concentration. Finally, the CMC value was calculated from the intersection of the tangent of the tangent and the horizontal tangent of the curve [23].

### 2.5. Preparation of RDP/I

The nanoheater RDP/I was prepared as follows: peptide and IR780 (*w*/*w* = 5) were dissolved in 5 mL of DMF. Then, 1 mL of water was slowly added to the above solution and stirred for 1 h. Afterwards, the solution was transferred to dialysis membrane filters (MWCO: 3500 Da) and dialyzed with distilled water for 72 h. Finally, the mixed solution in the dialysis membrane filters was centrifuged, and the supernatant was the RDP/I solution [24].

### 2.6. Characterization of RDP/I

The absorbance spectra of RDP/I and IR780 were detected by a UV-vis spectrophotometer (Lambda Bio40, Perkin-Elmer, MA, USA). The particle size and the zeta potential of RDP/I were analyzed by Nano-ZS ZEN3600 [25]. And the morphology of RDP/I was observed by transmission electron microscopy (TEM) (JEM-2100, JEOL, Akishima Japan) [26].

### 2.7. Release Behavior In Vitro

The dialysis membrane filter (MWCO: 3500 Da) containing 1 mL RDP/I was immersed in 10 mL buffer solution (pH 7.4, 6.5, or 5.0) and shaken at 37 °C for in vitro release simulation [27]. The encapsulation efficiency (EE) and loading level (LL) of IR780 are defined as below:EE = (mass of loaded IR780/mass of feed IR780) × 100%
LL = (mass of IR780/mass of RDP/I) × 100%

### 2.8. Photothermal Efficiency Tests

Different concentrations of RDP/I and IR780 solutions (the IR780 concentration was 15 μg/mL, 20 μg/mL, or 30 μg/mL) were irradiated with a NIR laser (LWIRL808, Laserwave, NJ, USA) at 808 nm at a power of 1 W for 1 min. The thermal images and the temperature of samples were observed regularly by a FLIR Ax5 camera (ONE LT, FLIR, OR, USA) [28]. The photothermal efficiency of samples (IR780 concentration of 30 μg/mL) that were irradiated with a NIR laser at 808 nm at different powers (0.75 W, 1 W, or 1.5 W) for 1 min was recorded by the same method.

### 2.9. Cytotoxicity Assay In Vitro

The cytotoxicity of RDP/I and IR780 in CT26 cells and NIH3T3 cells was evaluated via the MTT assay [29]. Cells (6000 cells/well) were seeded in 96-well plates and incubated for 24 h. Then, different concentrations of samples were added to the plates and incubated with cells. After incubating for 4 h, the medium was removed and 200 μL of fresh medium was added to each well. Afterwards, cells were irradiated with a NIR laser (808 nm, 1 W) for 1 min and then incubated for another 48 h. After that, 20 μL of MTT solution was added and cultured for 4 h. Finally, the medium was replaced with 200 μL of DMSO, and the absorbance at 490 nm was measured by a microplate reader (iMark, Bio-rad, CA, USA). The cytotoxicity without the NIR laser irradiation of samples was also investigated by the same procedure.

For live cell staining assays, CT26 cells were seeded in 6-well plates. After and cultured for 24 h. Then, RDP/I or IR780 were added to the plates. Cells incubated with PBS were used as the negative control. After incubating for 4 h, the medium was substituted with fresh medium, and cells were irradiated by a NIR laser (808 nm, 1 W) for 1 min or placed in the dark. After incubation for another 48 h, all cells were incubated with Calcein AM solution and observed by fluorescence microscope (IX51FL, Olympus, Shinjuku, Japan) [30].

### 2.10. In Vitro Wound-Healing Assay

After CT26 cells (1 × 10^5^ cells/well) incubated in 6-well plates for 24 h, a line of cells were removed by pipette tip. Afterwards, the medium was replaced with fresh medium containing samples (the IR780 concentration was 0.25 μg/mL). 12 h later, the healing rate was observed by fluorescence microscopy [31].

### 2.11. Intracellular Uptake of RDP/I

Cell uptake behavior of RDP/I in CT26 cells was observed by CLSM. After cells were cultured in plates for 24 h, fresh medium containing RDP/I or IR780 (the IR780 concentration was 0.25 μg/mL) was added and incubated with cells for 4 h. Afterwards, Hoechst 33,342 was added to label the cell nucleus, and the intracellular uptake behavior of samples was observed by a Confocal Laser Scanning Microscope (CLSM) (C1-Si, Nikon, Tokio, Japan) [32].

The quantitative analysis of the intracellular uptake behavior of samples in CT26 cells was assessed by flow cytometry. After being cultured with samples for 4 h, cells were dissociated and re-suspended in PBS. Finally, the fluorescence intensity of intracellular samples was quantitatively detected by flow cytometry (BD Accuri™ C6 Plus, BD, NJ, USA) [33].

### 2.12. Tumor Suppression Study

CT26 cells (2 × 10^4^ cells/well) were cultured in 6-well plates for 24 h. Then, the fresh medium containing RDP/I (IR780 concentration was 0.25 μg/mL) was added to plates and incubated with cells for 4 h. After being washed with PBS three times to remove samples, cells were irradiated with or without a NIR laser (808 nm, 1 W) for 1 min and incubated for another 48 h. Finally, cells were digested by trypsin (without EDTA) and re-suspended in annexin-binding buffer. After being stained with Annexin V-FITC and propidium iodide (PI), cells were detected by flow cytometry to evaluate their apoptosis and necrosis. Cells incubated with PBS were used as the negative control, while those incubated with IR780 under a NIR laser were used as the positive control [34].

Agarose solution (2.5%, *v*/*v*) was prepared and transferred to a 96-well plate (200 μL**/**well). After the solution was solidified, the cells (1 × 10^5^ cells/well) were evenly seeded in 96-well plates. The fresh medium was changed every other day to ensure the growth of cell mass. After 7 days, the medium was replaced with the medium containing samples. After 24 h, medium was substituted with fresh medium, and cells were irradiated with or without a NIR laser (808 nm, 1 W) for 1 min on the second and fourth days. Finally, continue to culture the cell mass for 7 days; the growth of cell clusters was observed regularly by microscope [35].

## 3. Results and Discussion

### 3.1. Characterization of RDP

Firstly, we synthesized RDP with the sequence CRRK(RRCG(Fmoc))_2_ via the standard solid-phase synthesis method. The molecular weight of CRRK(RRCG(Fmoc))_2_ should be calculated at 1949.92. Multiple molecular weights such as 391.2 ([M + 5H]^5+^/5), 488.5 ([M + 4H]^4+^/4), 650.8 ([M + 3H]^3+^/3), and 976.3 ([M + 2H]^2+^/2) can be found in ESI-MS; the results were consistent with expectations (Table 1).

To investigate whether the peptide CRRK(RRCG(Fmoc))_2_ can spontaneously form a micellar structure, the CMC of the peptide was assessed by fluorescence spectrophotometer, and pyrene was used as a probe. As shown in Figure 2A, with the increase in peptide concentration, the fluorescence intensity of pyrene increased. The CMC value of the peptide was calculated to be about 19.6 mg/L, indicating that when the concentration of peptide was higher than its CMC value, it could self-assemble to form micelles by hydrophilic and hydrophobic interactions. Using this characteristic, the hydrophobic photothermal agent IR780 could be encapsulated in its hydrophobic core to form nanomicelles RDP/I by self-assembly.

### 3.2. Characterization of RDP/I

Further, the performance of nanoheater RDP/I is evaluated through comprehensive characterization. The loading behaviors, including EE and LL, of IR780 in RDP/I were determined first. According to the calculation formula, the EE and LL values of RDP/I were calculated to be 28.8% and 9.6%, respectively. The encapsulation behavior indicated that RDP/I can effectively encapsulate the photothermal agent IR780 and has great potential as a carrier of photothermal agents for antitumor therapy.

Then, the absorption spectra of RDP/I and IR780 were analyzed by a UV-vis spectrophotometer. The surface plasmon resonance (SPR) peaks of RDP/I and IR780 are both located at 780 nm (Figure 2B). These data indicated that IR780 was successfully encapsulated in RDP/I, and both RDP/I and IR780 had good potential for NIR photothermal conversion therapy.

The morphology of RDP/I was observed by TEM, and it was a uniform spherical particle of about 20 nm in vacuum (Figure 2C). After that, the hydrated particle size of RDP/I was measured to be 137.2 nm by dynamic light scattering. The nanoheater, with particle sizes between 100 and 200 nm, was conducive to accumulation at the tumor site through the EPR effect. The zeta potential of RDP/I was +30.2 mV, which was available for cell internalization (Table 2).

In order to evaluate whether RDP/I could effectively encapsulate and deliver IR780 to the nucleus, the release behaviors of RDP/I in different intracellular pH environments were detected. As shown in Figure 2D, after 72 h of treatment, the release amounts of RDP/I at pH = 7.4, 6.5, and 5.0 were only 9.3%, 11.4%, and 19.9%, respectively. The results demonstrated that RDP/I can effectively encapsulate IR780 to prevent its release in different pH environments. The pH-independent, long-effective protection of IR780 by RDP/I provides a strong guarantee for nuclear delivery.

### 3.3. Photothermal Efficiency Tests

The photothermal conversion ability was evaluated by detecting the temperature rise of RDP/I, while that of PBS and IR780 were tested as the negative control and the positive control, respectively. With the extension of irradiation time, the temperature of the RDP/I solution increased rapidly; the ΔT of RDP/I was slightly lower than that of IR780, while PBS had no obvious photothermal effect (Figure 2E).

In addition, we further investigated the photothermal stability of RDP/I and IR780. In the three heating and cooling cycles (808 nm, 1 W), the RDP/I showed relatively constant temperature changes, and the peaks of ΔT in the three cycles were 23.5 °C, 19.3 °C, and 15.7 °C, respectively. Nevertheless, during the three cycles, the ΔT of IR780 decreased sharply. In the first cycle, the ΔT of IR780 heated up to 30.5 °C. While in the second cycle, the ΔT of IR780 greatly reduced to 12.8 °C; the ΔT further decreased to 5.7 °C in the third cycle (Figure 2F). These results confirm that the photothermal stability of RDP/I is much stronger than that of IR780.

The temperature rise of RDP/I at different concentrations or irradiation intensities was also measured (Figure 3). With the increasing concentration, the temperature of RDP/I also increased, the data indicating that the photothermal conversion ability of RDP/I is positively correlated with the concentration. When the irradiation power increased from 0.75 W to 1.5 W, ΔT increased from 20 °C to 28.4 °C at 60 s, indicating that the photothermal conversion ability of RDP/I depends on the laser power. The above results again verify the excellent photothermal conversion performance of RDP/I.

### 3.4. Cytotoxicity Assay In Vitro

Cytotoxicity assays in NIH3T3 and CT26 cells were performed to evaluate the therapeutic efficiency of samples. In this part of the experiments, RDP/I and IR780 with NIR irradiation were labeled as RDP/I+ and IR780+, while groups without NIR irradiation were labeled as RDP/I− and IR780−. As shown in Figure 4A,B, although the growth inhibition ability of IR780 was stronger than that of RDP/I in CT26 cells, the cell survival rate of IR780+ was only significantly different from that of IR780− in the narrow concentration range of 0.063~0.125 μg/mL. While in the wide concentration range from 0.008 μg/mL to 1.3 μg/mL, the cell survival rate of RDP/I− was much higher than that of RDP/I+. The IC50 values of IR780+, IR780−, RDP/I+, and RDP/I− in CT26 tumor cells were 0.063 μg/mL, 0.146 μg/mL, 0.251 μg/mL, and 0.665μg/mL, respectively. In NIH3T3 normal cells, the IC50 values of IR780+, IR780−, RDP/I+, and RDP/I− were 0.092 μg/mL, 0.146μg/mL, 0.375 μg/mL, and 0.983 μg/mL, respectively. These results indicate that RDP/I could improve antitumor activity in tumor cells and weaken the adverse effects in normal cells by precise localization through the regulation of NIR irradiation.

In the live cell staining experiment, only weak green fluorescence was observed in the IR780+ and IR780− groups, and the fluorescence difference between the two groups was negligible. While the fluorescence intensity between the RDP/I+ and RDP/I− groups was significantly different, the green fluorescence in the RDP/I+ group was much stronger than that of the RDP/I− group. The results implied that IR780 has strong cytotoxicity with or without NIR irradiation, while RDP/I can achieve controllable cell inhibition through NIR regulation.

### 3.5. In Vitro Wound-Healing Assay

The results of cell migration experiments were consistent with those of live cell staining experiments (Figure 4D). Only a few living cells and blurred wounds were observed in the IR780+ and IR780− groups, which showed strong cell toxicity of IR780. A clear wound could be found in the RDP/I+ and RDP/I− groups, and the wound in the RDP/I+ group was much wider than that in the RDP/I− and PBS groups. The data suggests RDP/I+ could effectively kill cells and inhibit cell migration. Moreover, the number of living cells in the RDP/I− group was closest to that in the PBS group. The above results again indicate that RDP/I can achieve efficient tumor inhibition and effectively reduce non-target side effects through NIR regulation.

### 3.6. Intracellular Uptake ofRDP/I

In order to track the nuclear delivery behavior of nanoplatform RDP/I, the intracellular distribution of samples was observed by CLSM. As shown in Figure 5A, after co-culture with cells for 4h, both IR780 and RDP/I showed red fluorescence in the cells. The red fluorescence of the IR780 group was stronger than that of the RDP/I group, but the red fluorescence of the RDP/I group overlapped with the blue fluorescence of the nucleus in a large area, indicating that RDP/I was successfully localized to the nucleus. The above results identified that RDP/I has excellent nuclear localization ability, which is conducive to precise phototherapy of the nucleus. The fluorescence intensity of intracellular samples was further quantitatively detected by flow cytometry. As shown in Figure 5B,C, the intracellular fluorescence intensity of the RDP/I group can reach the same order of magnitude as IR780, which proved that RDP/I can achieve efficient cell internalization.

### 3.7. Tumor Suppression Study

The apoptosis-inducing ability of RDP/I on tumor cells was verified by flow cytometry. The total amount of apoptosis and necrosis in the RDP/I− group was 14.5%, while that of the RDP/I+ group was 62.9%. The apoptosis-inducing ability of RDP/I with NIR irradiation was significantly higher than that in the dark group. This result suggests that the photothermal effect produced by RDP/I with NIR irradiation has a significant killing effect on tumor cells (Figure 6A).

Further tumor inhibition studies were performed in 3D cell spheroids. When the diameter of cell spheres reached 300 μm, samples were added and co-cultured with cell spheres for 24 h (the first day). After that, replace the samples medium with fresh medium. NIR was irradiated at the second and fourth days. As shown in Figure 6C, the cell sphere diameter of the RDP/I group was close to that of the IR780 group; all of them were much smaller than that of the PBS group, and the cell sphere diameter of samples with NIR irradiation was smaller than that without NIR irradiation. All the data indicated that RDP/I with NIR irradiation could effectively inhibit the growth of tumors.

## 4. Conclusions

In this study, the nanoplatform RDP/I for precise nucleus-targeting phototherapy was prepared by encapsulating the photothermal agent IR780 with an arginine-rich amphiphilic dendritic peptide. The nanoplatform successfully delivered the photothermal agent to the nucleus and achieved precise photothermal therapy of the tumor nucleus. The hydrophobic group Fmoc of the amphiphilic peptide provided strong hydrophobic force to firmly encapsulate IR780, which improved its solubility and stability. Due to the membrane permeability and nuclear localization function of peptide, the nanoplatform RDP/I was delivered to the nucleus efficiently and achieved efficient tumor inhibition efficiency by precise nucleus-targeting photothermal therapy. More importantly, the nanoplatform can reduce the side effects of non-targets by regulating NIR. The construction of this nanoheater for precise delivery of nuclei provided a simple and efficient strategy for future precise treatment.

## Figures and Tables

**Figure 1 polymers-15-01753-f001:**
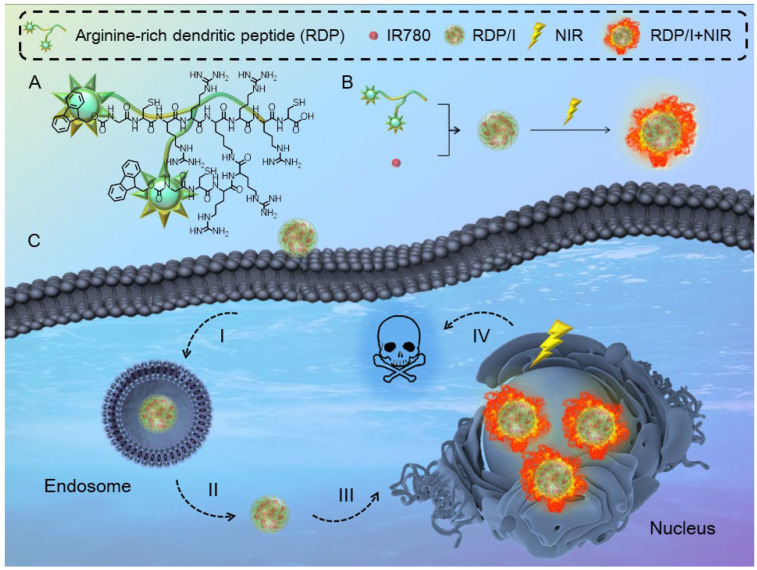
(A) Structural formula of RDP; (B) The synthesis diagram of RDP/I; (C) Schematic diagram of the nanoheater RDP/I for precise nucleus-targeting photothermal therapy, (I) Nanoheater RDP/I was internalized in cells. (II) RDP/I escaped from the endosome. (III) RDP/I was anchored to the nucleus. (IV) Cell death due to the precise nuclear photothermal effect of RDP/I.

**Figure 2 polymers-15-01753-f002:**
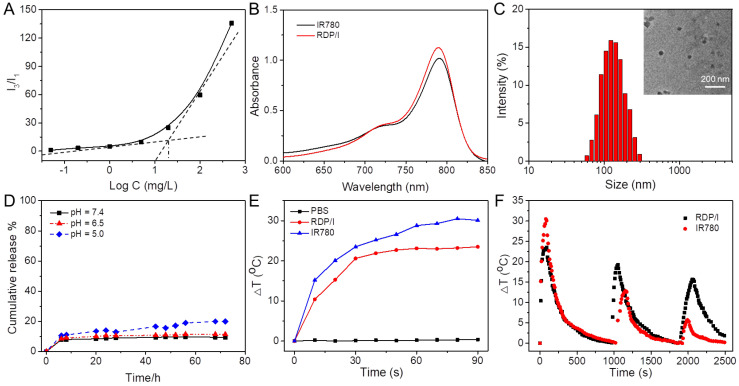
(**A**) The CMC of RDP; (**B**) UV-vis absorption spectra of RDP/I and IR780; (**C**) particle size and TEM image of RDP/I; (**D**) release behavior of RDP/I; (**E**) temperature changes of samples (IR780 concentration was 30 μg/mL) with NIR irradiation (808 nm, 1 W); (**F**) cyclic stability test of samples (IR780 concentration was 30 μg/mL) with NIR irradiation (808 nm, 1 W).

**Figure 3 polymers-15-01753-f003:**
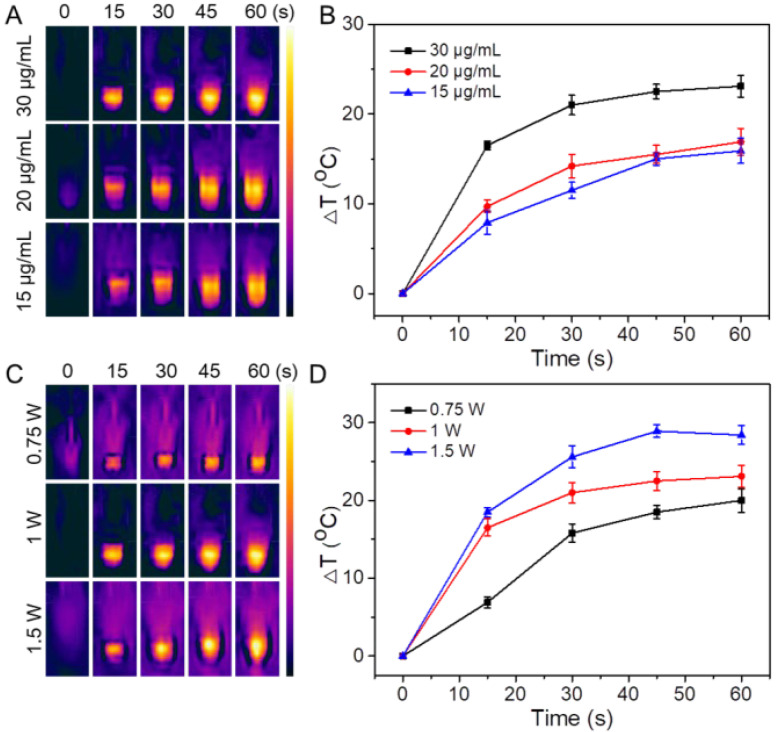
(**A**) Photothermal images and (**B**) the heating curve of RDP/I at different concentrations with NIR irradiation (1 W); (**C**) photothermal images and (**D**) the heating curve of RDP/I (IR780 concentration was 30 μg/mL) with NIR irradiation at different laser powers.

**Figure 4 polymers-15-01753-f004:**
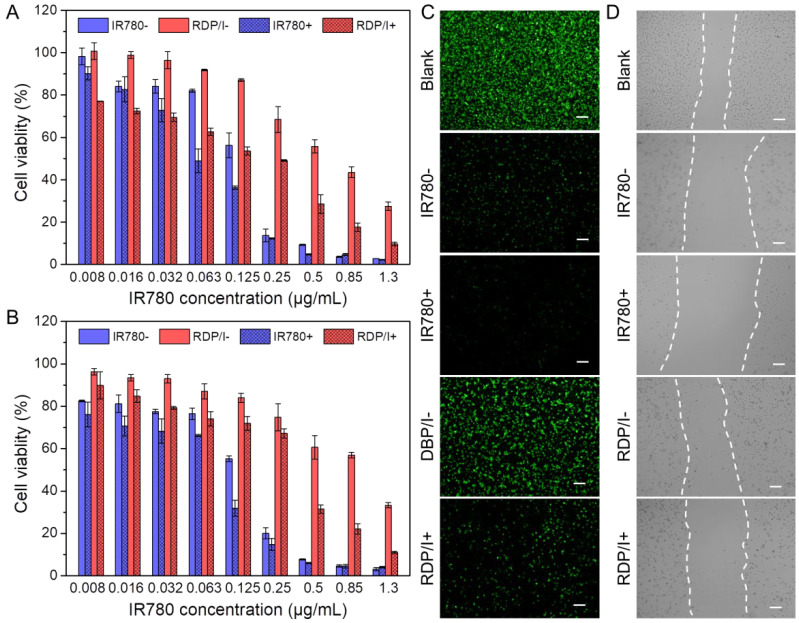
In vitro anti-tumor performance of samples. Cytotoxicity of samples against (**A**) CT26 cancer cells and (**B**) NIH3T3 normal cells; data are shown as mean ± SD (n = 4); (**C**) fluorescent images of sample treated CT26 cells stained with Calcein AM; (**D**) wound healing assay of sample treated CT26 cells; the scale bar: 50 μm.

**Figure 5 polymers-15-01753-f005:**
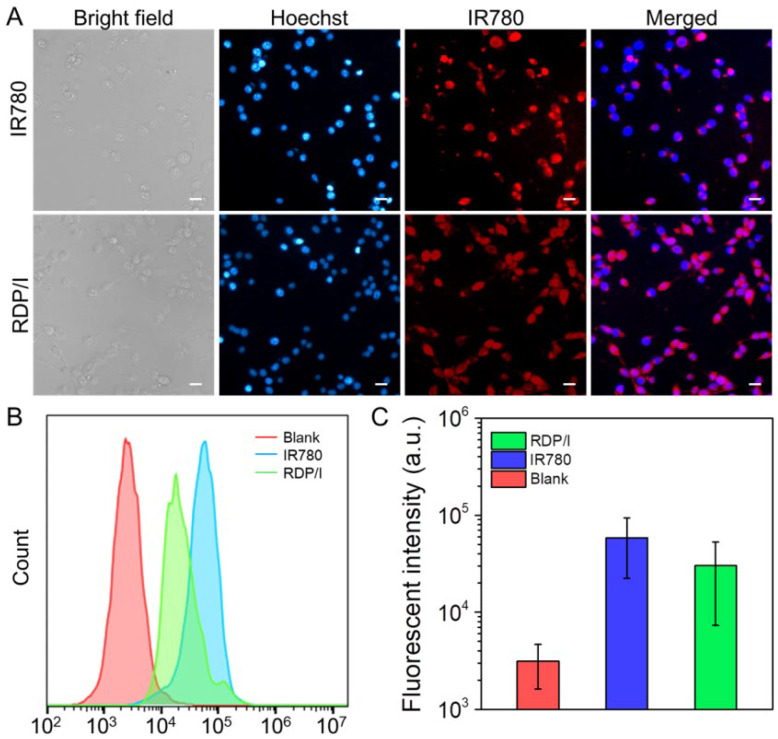
(**A**) CLSM images of CT26 cells after being treated with samples for 4 h; scale bar: 20 μm; (**B**,**C**) quantitative analysis of intracellular fluorescence intensity in CT26 cells after being treated with samples for 4 h.

**Figure 6 polymers-15-01753-f006:**
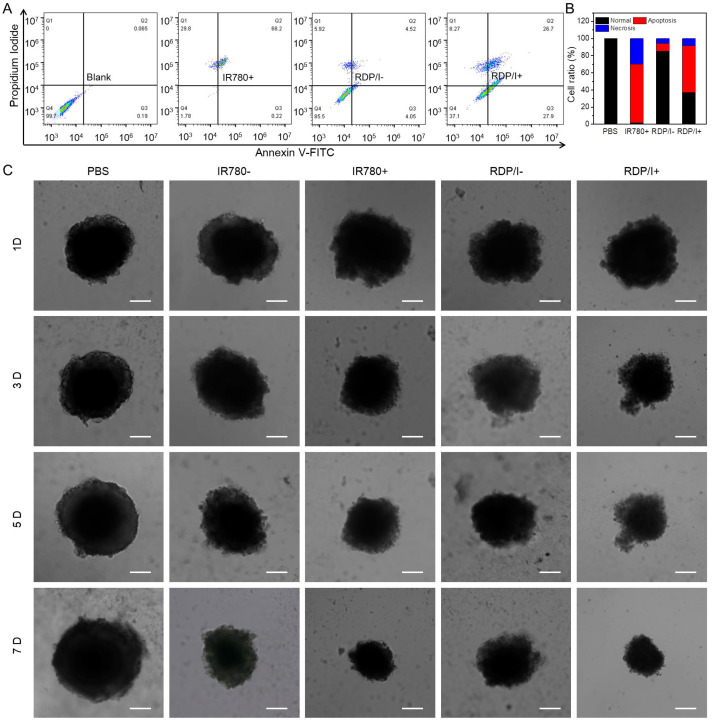
(**A**,**B**) Quantitative measurement of apoptosis by AnnexinV-FITC/PI in CT26 cells after treated with samples in the absent or present of NIR irradiation (808 nm, 1 W, 1 min); (**C**) growth inhibition of CT26 cells spheroids after treated with samples in the absent or present of NIR irradiation (808 nm, 1 W, 1 min); scale bar: 100 μm.

**Table 1 polymers-15-01753-t001:** ESI-MS analysis of peptide.

Peptide	M (Calculated)	m/z (Found)
CRRK(RRCG(Fmoc))_2_	1949.92	[M + 5H]^5+^/5: 391.2 [M + 4H]^4+^/4: 488.5 [M + 3H]^3+^/3: 650.8 [M + 2H]^2+^/2: 976.3

**Table 2 polymers-15-01753-t002:** Hydrodynamic diameter and zeta potential of RDP/I.

Sample	Hydrodynamic Diameter (nm)	Zeta Potential (mV)
RDP/I	137.2 ± 0.6	+30.2 ± 0.4

## Data Availability

The data presented in this study are available in this study. Additional information could be available upon request from the corresponding author.

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
