# Peer review of "Nucleus-Targeting Nanoplatform Based on Dendritic Peptide for Precise Photothermal Therapy"

_polymers, 2023, doi:10.3390/polym15071753_

Round 1
Reviewer 1 Report
The manuscript 'Nucleus-targeting nanoplatform based on dendritic peptide for precise photothermal therapy' by Wen-Song Wang et al reports on functional tests of micelles based on arginine-rich branched peptide CRRK(RRCG(Fmoc))2 modified with IR780 as a novel and efficient medium for the precise nucleus-targeting photothermal therapy of tumor. The manuscript is suitable for the special issue Protein-Based Biopolymers of Polymers. The manuscript is generally well written with clear and well argumented conclusions. I recommend the authors to consider a few minor corrections.
1. RPMI should read Roswell Park Memorial Institute medium.
2. Device models, manufacturers, operation modes and parameters should be briefly given for TEM, UV-Vis, flow cytometer, spectrofluorimeter, etc.
3. Essential details of the synthesis of the Dendritic peptide CRRK(RRCG(Fmoc))2 should be given. Please, check the reference quoted. Probably, [24] should be instead of [22].
4. Please, comment how the molecular weight of 1949.92 a.m.u. should follow from the chemical notation CRRK(RRCG(Fmoc))2 (please, indicate the amino acid composition and number of residues).
Reviewer 2 Report
The presented results are interesting and the quality of the performed experiments is good in my opinion. I would like to recommend to review the introduction and references. I am not sure that the broad audience of "polymers" is familiar with the properties of arginin-reach branched peptides, which you call RBP. Please extend the introduction by some sentences and give references for some properties. Additionally, I ask you to check your list of references. Not only Chinese scientists work in this field. The abbreviation RBP is also used for "receptor-binding protein" and could mislead the reader.
Author Response
Thank you for your suggestion. The introduction of dendritic peptides with arginine-rich sequences is supplemented in the manuscript.
The polymers which called “arginine-rich branched peptides (RBP)” was changed to “arginine-rich dendritic peptides (RDP)”. The abbreviations of RDP were explained in abstracts, prefaces, schematic and abbreviation list to avoid readers' misunderstanding.
Reviewer 3 Report
This manuscript introduced a nanoplatform RBP/I for precise nucleus-targeting phototherapy which was prepared by encapsulating photothermal agent IR780 with arginine-rich amphiphilic branched peptide. The nanoplatform successfully delivered the photothermal agent to nucleus and achieved precise photothermal therapy of tumor nucleus.
The idea is novel and the experiments are well designed.
However, the English language of the manuscript needs a thorough revision where many grammatical and punctuatuion mistakes exist. For example the past tense of " lead " is " led " not " leaded " and so on ...
Also, in section 3.3, the authors should elaborate how was the photothermal efficiency test was performed ? What was the equipment used for generating the NIR laser at 808 nm ?
Performing the cytotoxicity assays under NIR irradiation to detect the photothermal activities of compounds or nano structures was previously used in: https://doi.org/10.1016/j.jddst.2018.07.002 and International Journal of Nanomedicine, 2605-2615 that should be mentioned in the text.
The authors should add the error bars in figure 3 and determine its nature
Author Response
Many thanks for your comments of our manuscript submitted to Polymers (Manuscript ID: polymers-2282344). The answers to comments were listed below.
1. The idea is novel and the experiments are well designed. However, the English language of the manuscript needs a thorough revision where many grammatical and punctuatuion mistakes exist. For example the past tense of "lead" is " led " not " leaded " and so on ...
Reply: According to the suggestion, we have carefully revised the manuscript.
2. Also, in section 3.3, the authors should elaborate how was the photothermal efficiency test was performed ? What was the equipment used for generating the NIR laser at 808 nm ?
Reply: Thank you for your suggestion. The experimental procedure of the photothermal efficiency test is supplemented in the manuscript. Device model and parameter of NIR laser was also supplemented accordingly.
3. Performing the cytotoxicity assays under NIR irradiation to detect the photothermal activities of compounds or nano structures was previously used in: https://doi.org/10.1016/j.jddst.2018.07.002 and International Journal of Nanomedicine, 2605-2615 that should be mentioned in the text.
Reply: Revised as suggested (reference 29).
4. The authors should add the error bars in figure 3 and determine its nature.
Reply: Revised as suggested (Figure 3).